# Validity and Reliability of Isometric-Bench for Knee Isometric Assessment

**DOI:** 10.3390/ijerph17124326

**Published:** 2020-06-17

**Authors:** Johnny Padulo, Nebojša Trajković, Drazen Cular, Zoran Grgantov, Dejan M. Madić, Rosa Di Vico, Alfonso Traficante, Larion Alin, Luca Paolo Ardigò, Luca Russo

**Affiliations:** 1Department of Biomedical Sciences for Health, Università degli Studi di Milano, 20133 Milan, Italy; sportcinetic@gmail.com; 2Faculty of Sport and Physical Education, University of Novi Sad, 21000 Novi Sad, Serbia; nele_trajce@yahoo.com (N.T.); dekimadic@gmail.com (D.M.M.); 3Faculty of Kinesiology, University of Split, 21000 Split, Croatia; dcular@kifst.hr (D.C.); zoran.grgantov@gmail.com (Z.G.); 4“Einstein” Craft for Research, Development, Education, Trade and Services, 21000 Split, Croatia; 5Italian Society of Posture and Gait Research, 8100 Caserta, Italy; rosadivico@gmail.com; 6Department of Biotechnology and Applied Clinical Science, University of L’Aquila, 67100 L’Aquila, Italy; fonz91@hotmail.com (A.T.); luca83russo@gmail.com (L.R.); 7Faculty of Physical Education, Ovidius University of Constanta, 900029 Constanta, Romania; alinlarion@yahoo.com; 8School of Exercise and Sport Science, Department of Neurosciences, Biomedicine and Movement Sciences, University of Verona, 37131 Verona, Italy

**Keywords:** concurrent validity and reliability, field test, isometric assessment, testing

## Abstract

There is a strong need for a new, probably cheaper, smaller, and more portable isometric dynamometer. With this aim, we investigated the concurrent validity and reliability of a low-cost portable dynamometer to measure the isometric strength of the lower limb. Seventeen young participants (age 16.47 ± 0.51 years) were randomly assessed on three different days for knee flexion and extension isometric forces with two different devices: a commonly used isokinetic dynamometer (ISOC) and a portable isometric dynamometer prototype (ISOM). No significant differences were observed between the ISOC and the ISOM (all comparisons *p* > 0.05). Test–retest comparison showed the ISOM to have high reliability (ICC 0.879–0.990). This study showed that measurements with the ISOM could be performed without systematic bias and with high reliability. The ISOM is a device that is able to assess knee isometric strength with excellent concurrent validity and reliability.

## 1. Introduction

Sufficient muscle strength is crucial for both successful completion of daily activities such as standing up from a chair or getting out of bed [1] alongside other pathological subjects [2]. Isometric strength was found to be beneficial for several sports, such as football and wrestling [3], but also for track cycling [4] and track and field [5]. Moreover, McGuigan et al. [6] found that the mid-thigh isometric pull is strongly related to the performance of other dynamic tests such as the vertical jump and the one-repetition *maximum* [7]. Supported by those findings, peak isometric strength assessment was used to determine effects of training—such as whole-body vibration and stretching—on performance [8,9]. Moreover, a deficiency in isometric strength has been considered as a risk factor predicting the need for knee replacement surgery in women [10].

Therefore, monitoring muscle strength is a relevant procedure in minimizing potential risk [11]. Moreover, the results of strength testing after a knee injury are of great importance to clinicians when deciding to return to function and when authorizing participation in physical activity [12]. Muscle strength assessment is therefore a crucial point in both clinical and research settings, allowing the tracking of rehabilitation progress and exercise interventions to be designed accordingly. Assessing isometric muscle strength with custom-built load cells has been reported in the literature [13,14,15,16]. There are several off-the-shelf devices which assess muscle strength in both upper and lower limbs [17]. Isokinetic dynamometry represents the gold standard of assessing flexion and/or extension muscle strength.

Dynamometers such as the not-portable-with-seat one used in this study (Biodex System 3 pro, Shirley, NY, USA; ISOC) provide reliable measurements, particularly of lower extremity muscle strength, and are considered the gold standard of muscle strength measurement tools [18]. However, the ISOC is expensive (USD 30,000), requires extensive familiarization, lacks portability, and takes up a substantial amount of clinical space. Nevertheless, maximum voluntary contraction to assess muscle strength related to knee flexion and extension is commonly used in both clinical and sport science with some limitations. On the contrary, the portable-with-seat dynamometer prototype (Isometric-Bench, Rome, Italy; ISOM; Figure 1) assessed in this study is a newly customized device that measures the maximum isometric force developed by the quadriceps femoris and biceps femoris muscles. The validity and reliability of this device compared with a gold standard strength assessment tool have not yet been reported in the literature. Therefore, the aim of this study was to investigate the concurrent validity and reliability of a probably low-cost, portable dynamometer prototype to assess knee flexion and extension isometric strength.

## 2. Materials and Methods

Seventeen healthy males without history of orthopedic diseases, aged 16.47 ± 0.51 yrs, with a body mass of 71.8 ± 9.45 kg (57–93 kg), and a height of 1.75 ± 0.06 m (1.67–1.90 m) voluntarily participated in the study. They had experience in football (10.29 ± 1.05 yrs). They had a weekly practice of 10 hours during the national phase of the championship (Italian U17, Serie C, Girone D). All participants had the following features: they participated in at least 85% of the training sessions, possessed a valid sport medical certification, had suffered no injuries in the last year, and had no consumption of alcohol or drugs. Participation was voluntary and written and verbal information was provided to all participants. Moreover, they were informed in detail about the aims, benefits, and potential risks of the study. Informed consent was obtained from the participants and their guardians prior to the experiments. Ethical approval was issued by the local university ethics committee (University of Novi Sad Ethics Committee; protocol number: 1/2019; 06/03/2019) following ethical standards for human studies after having approved all experimental procedures according to the Declaration of Helsinki Code of Ethics (Statement of Helsinki).

Prior to the test, study participants abstained from consuming alcohol or caffeinated beverages for 24 hours and did not consume food for 3 hours to reduce any interference. Each participant completed all the tests during the same period of the day and under the same environmental conditions (3:00–5:00 p.m., 21.5 ± 0.3 °C, and 46.2 ± 1.4% relative humidity) to avoid the influence of any circadian variation. All tests were performed in a rehabilitation center (nursing home ‘San Michele’, Maddaloni, Caserta, Italy) and participants wore sports clothing. Every participant was tested three times on three different days: on the ISOC, first time on the ISOM, and second time on the ISOM. The tests were administered randomly over the three different days. Anthropometric data, mass, height, and lower leg length (measured from the knee lateral joint line to the lateral malleolus) were measured and recorded for each participant. Prior to testing, all participants warmed up (10 min of self-selected low-intensity running) with the help of two assistants.

Knee flexion and extension isometric tests with the ISOC and the ISOM were administered in a randomized order. Knee flexion and extension forces were assessed with a 90° knee flexion. The joint angle was calibrated by means of software (Biopac Systems, Goleta, CA, USA) and an electronic goniometer (MuscleLab, Bosco System, Langesund, Norway) for the ISOC and the ISOM, respectively. With both devices, particular care was taken to block the knee to effectively avoid its movement. For every randomly chosen condition (right leg flexion, left leg flexion, right leg extension, and left leg extension), the settings of the dynamometers were adapted differently. Particular care was taken to maintain the participant’s posture (and the resulting knee moment arms) with both dynamometers. The participant was firmly secured to the seat and tried to relax maximally. Following the operator “go”, participant used maximum effort to flex (or extend) the knee for four sec. Each test was repeated if the participant started flexion (or extension) before the operator order. For each condition, two measurements were taken and the best (viz. strongest) result was used for subsequent analysis.

In this study, the ISOC and the ISOM were randomly used to assess the isometric strength of knee flexors and extensors. The commonly used isokinetic dynamometer chosen in this study for knee flexion and extension assessment represents the gold standard of isometric evaluation [19]. The ISOC was calibrated prior to the testing of each participant following the manufacturer manual. The knee lateral epicondyle was aligned with the dynamometer rotation axis and the participant was blocked with straps. The lower leg attachment was adjusted to 3 cm above the malleolus and was fixed using a non-elastic band to limit its movements. Mechanical ISOC signals were recorded, digitalized, and sent to a computer with a 16-bit A/D data acquisition and analysis conducted with a 2-kHz sampling rate system (MP 150, Biopac Systems, Goleta, CA, USA) using the included software (Acq*Knowledge*, Biopac Systems, Goleta, CA, USA).

### Portable Dynamometer

The portable-with-seat dynamometer prototype (Isometric-Bench, Rome, Italy; ISOM; Figure 1) assessed in this study has a metallic steel structure, a maximum size of about 110 cm, and an assembly time of about 30 min (Figure 1). The structure’s main purpose is to fix the lower limb to assess knee flexion or extension isometric force with a load cell (Figure 1). The structure is provided with a seat to allow the user to sit and lock the hands to handles placed laterally. The device has a removable custom bracket (Element A, Figure 1) that can be placed at five different vertical heights (from seat height down to about 70 cm below that, for different lower limbs lengths), different antero-posterior positions (continuous regulation from the seat’s forward edge with a vertical projection up to 13 cm forward, for different knee angles), and the ability to point backwards (for flexion) or forwards (for extension). Element A was differently placed vertically and horizontally according to each participant’s shank length. A load cell (750 kg, 1000 Hz; H3-C3/C4-750kg-3B, Zemic Europe, Etten Leur, Netherland) is connected (i.e., embedded in) to Element A, which in turn connects the load cell to a metal plate perfectly in contact with the tested lower limb skin (pressure point). Namely, the load cell measures the horizontal (or “tangential”) knee flexion or extension isometric force applied against it. The structure is completed by a support plate for the foot, where the foot was softly placed on without weighing significantly on it. The foot support plate can be placed at any of a number of different vertical heights (from seat height down to about 80 cm below that, for different lower limb lengths). The pressure point is chosen on the shank skin, at two thirds distance from knee, and just above the ankle. By adjusting the ISOM settings differently, it is possible to assess ankle or elbow flexion or extension isometric force. Due to the novelty of the ISOM, it took an appropriate time to study its manual and practice enough with the ISOM before starting the testing of participants. Author LR adjusted the settings (set-up time about 5 min), RD took measures (including anthropometric ones), and JP supervised all operations. Mechanical ISOM signals were recorded and digitalized by a portable device (5 kN full scale; 1 N reading graduation; 24-bit resolution; FC5k, Axis, Gdańsk, Poland).

Data were presented in terms of averages and standard deviations. The ISOM strain gauge-provided force values in kN were converted into N (i.e., N = kN × 1000). The ISOM validity was analyzed with the Bland–Altman test to quantify measurement difference between the ISOM prototype and the ISOC gold standard [20]. The ISOM reliability was analyzed with the intraclass correlation coefficient (ICC [21]). The significance level was set at *p* ≤ 0.05. Statistical analysis (two-tailed paired samples Student *t*-test, *n* = 17 for each of the four sides/joints movements considered) was performed using SPSS Version 17 (IBM, Chicago, IL, USA).

## 3. Results

### 3.1. Validity

No significant differences were observed between the peak isometric forces performed using the ISOC and the ISOM (*p* > 0.05 for all comparisons; Table 1 including ISOC-ISOM difference 95% limits of agreement). The ISOC’s left and right knee extension peak isometric forces resulted (average ± standard deviation) in 415.40 ± 115.15 N and 402.81 ± 41.73 N respectively, compared to the corresponding ISOM’s values of 394.73 ± 123.28 N and 379.46 ± 50.78 N (+20.64 N/+5.1% and +23.39 N + 5.9%). The ISOC’s left and right knee flexion peak isometric forces resulted in 169.95 ± 44.09 N and 175.65 ± 31.60 N, respectively, compared to the corresponding ISOM’s values of 163.41 ± 46.72 and 170.29 ± 34.29 N (+6.54 N/+3.9% and +5.36 N/+3.1%). Therefore, only a small (i.e., 3–6%; Table 1) difference between measurements with the ISOC and measurements with the ISOM was observed, as illustrated in the peak isometric forces Bland–Altman plots as well (Figure 2). Most values fell within the 95% limits of agreement (LOAs, ±1.96 SDs) and there were no over- or under-estimation patterns noticed for either extension or flexion.

### 3.2. Reliability

The ISOM test-retest showed high reliability: 0.990 for left knee extension peak isometric force, 0.879 for right knee extension peak isometric force, 0.974 for left knee flexion peak isometric force, and 0.964 for right knee flexion peak isometric force.

## 4. Discussion

A portable, probably cheap, valid, and reliable system to objectively assess lower limb isometric strength could provide several populations with relevant benefits. This study showed that a portable-with-seat dynamometer prototype (ISOM) provides measurements without significant systematic bias when compared to a validated commercial not-portable-with-seat dynamometer (ISOC) and with high reliability. To investigate the validity of the ISOM, it was assessed concurrently with both devices.

This study’s comparison between the ISOC gold standard and the ISOM did not disclose any apparent substantial inter-device differences regarding their validity or reliability for any isometric strength measurements. According to the Bland–Altman plots, most values fell within the LOAs and no over- or under-estimation patterns were found for either extension or flexion. Similarly, regarding isometric strength assessment, Toonstra and Mattacola [12] found similar peak torque values for knee flexion and extension using an isokinetic dynamometer and a portable fixed dynamometer. Differently, they found significant differences between the isokinetic dynamometer and a third handheld dynamometer. Likewise, Reinking et al. [22] found significant differences between an isokinetic dynamometer and a handheld one while assessing knee extensors. However, according to Stark et al. [23], knee extensors are a muscle group which are able to develop a wide force range and thus require expert operators for accurate testing. This was confirmed by this study, where the LOA ranged from −24.26 to +71.04 N for right knee extension (Table 1). Regarding validity analysis, this study revealed disclosed small systematic inter-device biases. As systematic biases were small in most cases, the prototype showed excellent concurrent validity when assessing the isometric strength of the main knee movements.

Knee extension isometric force has widely been used as a knee joint strength outcome [22]. In particular, isokinetic dynamometry has previously been shown to be highly reliable in assessing isometric knee strength [22,24]. This study’s reliability measurements demonstrated good-to-high test-retest reliability for the ISOM for knee flexion and extension isometric strength assessment. This study shows that the investigated prototype provides reliable results for the assessment of the isometric strength of knee flexors and extensors and this study supports its use.

This study proposes a new, cheaper, smaller, and more portable device aiding commonly used isokinetic dynamometry due to the ISOM’s strong validity and reliability found in the study for knee flexion and extension isometric strength assessment. The ISOM takes up relatively little space, has a short set-up time, and could therefore show its usefulness and be widely used among sports professionals.

As a potential limitation of this study, it should be noticed that the small sample size investigated could have influenced the variability of strength values across participants. The variability of strength values particularly affects studies like this one, showing the absence of differences between measurements taken with different devices. Furthermore, the sample is about 16 years old, whereas there may be age-related differences. Moreover, findings may not be directly generalizable to some clinical populations, bearing in mind the fact that it is likely that muscle strength and power assessment would be particularly relevant in people with idiopathic or symptomatic muscle weakness [10]. Besides, isometric torques measurements were not explicitly considered. Dominant vs. non-dominant side measurements were not investigated. Furthermore, the calibrations of the dynamometers were not considered, but may affect the comparison of the devices. The complexity, duration, and cost of calibration are factors to be taken into consideration and should be matter for further studies. Finally, electromyography activity featuring the ISOM and the ISOC use was not investigated within this research and it should be taken into consideration in particular for future clinical studies. Nevertheless, despite these limitations, excellent validity and test-retest reliability were found for knee flexion and extension assessment with the ISOM. Overall, the authors believe that the ISOM represents an effective device for sports professionals aiming at assessing knee isometric force.

## 5. Conclusions

This study shows the low-cost dynamometer investigated to be a device with excellent concurrent validity and reliability for the assessment of isometric strength for the main knee movements. This new equipment could be used in a field situation due to its portability. Net of the study’s limitations acknowledged above, sports professionals could therefore consider the possibility of using this dynamometer to assess isometric strength even during the training process under field conditions.

The ISOM showed good validity compared to the ISOC. Moreover, the ISOM’s reliability for assessing knee muscle strength resulted good for both knee flexors and extensors. The ISOM is a simple evaluation tool that can be used in exercise and clinical settings to measure knee isometric muscle strength in young, male, healthy adults. Furthermore, the ISOM’s specific pros are cheapness and portability. This could allow the device to be used in a variety of strength and conditioning settings to assess training interventions or detraining effects, fatigue impact, readiness for competition, and after injury return to function.

This study also showed that, with the ISOM, valuable data could be easily collected and in a time efficient way, which would make such a device an acceptable alternative to the ISOC for practitioners aiming at measuring and monitoring muscle strength developed during muscular voluntary contraction.

## 6. Patents

Johnny Padulo and Luca Russo provided the portable isometric dynamometer prototype used for this study (patent number 202019000001440).

## Figures and Tables

**Figure 1 ijerph-17-04326-f001:**
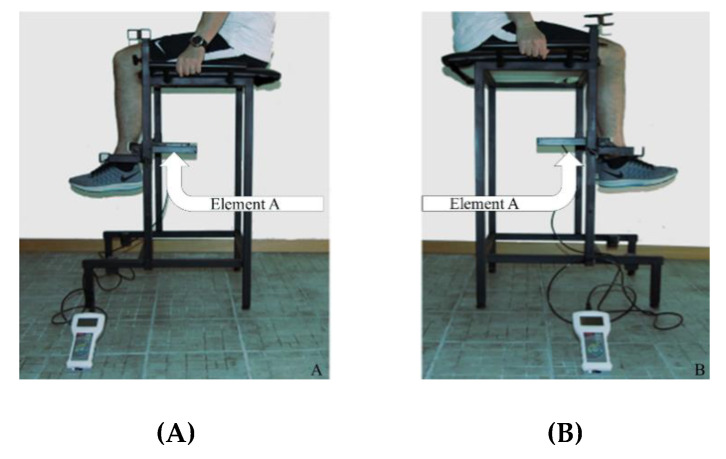
Isometric-Bench: flexion setting for left (**A**) and right (**B**) lower limbs.

**Figure 2 ijerph-17-04326-f002:**
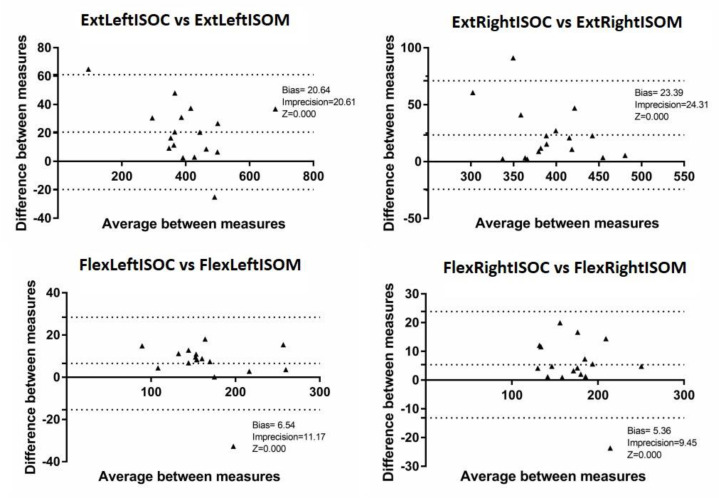
Knee extension and flexion peak isometric forces (N) Bland–Altman plots with both devices (*n* = 17). ExtLeftISOC = left knee extension with ISOC, ExtLeftISOM = left knee extension with ISOM, ExtRightISOC = right knee extension with ISOC, ExtRightISOM = right knee extension with ISOM, FlexLeftISOC = left knee flexion with ISOC, FlexLeftISOM = left knee flexion with ISOM, FlexRightISOC = right knee flexion with ISOC, FlexRightISOM = right knee flexion with ISOM.

**Table 1 ijerph-17-04326-t001:** Knee extension and flexion peak isometric forces with both devices (*n* = 17).

Variables	ISOC	ISOM	Sig	diff %	LOA
Left knee extension (N)	415.40 ± 115.15	394.73 ± 123.28	0.62	5.1	−19.76/+61.03
Right knee extension (N)	402.81 ± 41.73	379.46 ± 50.78	0.14	5.9	−24.26/+71.04
Left knee flexion (N)	169.95 ± 44.09	163.41 ± 46.72	0.68	3.9	−15.35/+28.43
Right knee flexion (N)	175.65 ± 31.60	170.29 ± 34.29	0.64	3.1	−13.14/+23.86

Data are presented as an average ± standard deviation for both devices (ISOC and ISOM). Sig = two-tailed paired samples Student *t*-test significance, LOA = ISOC–ISOM difference 95% limits of agreement (N).

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
