# Peer review of "Validity and Reliability of Isometric-Bench for Knee Isometric Assessment"

_ijerph, 2020, doi:10.3390/ijerph17124326_

Round 1

Reviewer 1 Report

  The paper presents an experimental study regarding the viability of a novel portable Isometric-Bench for isometric testing of the knee. The authors compare the prototype with a well-established, commercial device. Their results, obtained on a group of 17 young football players, show that the aformentioned prototype offers viable results when compared to a commercial device.
  My main concern with the paper is that it focuses too much on the experiment. The experiment only compares two biomedical devices - its novelty is limited. In my opinion, the main novelty lies in the concept of a portable prototype for isometric testing and this should be more clear in the paper. As of this version, the prototype is only briefly mentioned in the paper - it does not even have its own section. Furthermore, the device is vaguely cited, while its patent is not easily available - I was not able to find more information on it. This limits the reproducibility of the study and its possible applications.
  From what I deduced, the Isometric-Bench is a prototype, which has been patented by the first and the last author. I do not specialize in patents and I do understand that the authors are limited as to what they can include regarding the device. Nevertheless, the paper should at least have a separate section, which focuses on the structure and the novelties of the prototype. This way, the experiment would actually serve as a validation for the device, making the paper more complete in the process.
  I agree with the authors that portability and cost of dynamometers are very important and relevant issues. The prototype in the paper seems interesting and the experiment hints at its viability. Nevertheless, in my opinion, the paper should be restructured and partially rewritten so that it presents some novel aspects of the prototype and the experiment serves as a validation for it.

Other issues:
- the paper includes too many self-citations -> 13 out of 32 references include the name of the first author; in my opinion, the percentage of self-citations should not normally exceed 20%, currently it is at 40% - only the most important references should remain in the paper,

- [L188-L194] this paragraph does not belong in the discussion, it is more suited for the introduction,

- [L195] the study did not confirm ISOC as gold standard - this was the initial assumption,

- [L198-L200] in this section ISOC and ISOM were compared in terms of the design - it would be acceptable if more focus was put on the design of ISOM, as I mentioned in the beginning of the review,

- [L210] "probably" is a little awkward,

- in general the quality of language is sufficient, but some sentences could be rewritten (to make the paper easier to read).

Author Response

Response to Reviewer 1 Comments

English language and style

(x) English language and style are fine/minor spell check required

English language and style were professionally revised by an academic proofreading service.

Are the methods adequately described?

(x) Must be improved

Please, read below comments to specific points.

Point 1: As of this version, the prototype is only briefly mentioned in the paper - it does not even have its own section.

Response 1: We thank expert reviewer for his/her suggestion. An own prototype section was added. Its title is “2.1. Portable dynamometer” and starts as follows:

“Here assessed portable-with-seat dynamometer prototype (Isometric-Bench, Rome, Italy; ISOM; Figure 1) has a metallic steel structure, maximum size about 110 cm, and assembly time about 30 min (Figures 1 and 2).”

Point 2: Furthermore, the device is vaguely cited, while its patent is not easily available - I was not able to find more information on it.

Response 2: Patent is pending. This was specified as follows:

“Johnny Padulo and Luca Russo provided the portable isokinetic dynamometer prototype used for present study (patent pending 202019000001440).”

Point 3: the paper includes too many self-citations -> 13 out of 32 references include the name of the first author; in my opinion, the percentage of self-citations should not normally exceed 20%, currently it is at 40% - only the most important references should remain in the paper.

Response 3: First author’s self-citation rate was decreased down to 9%.

Point 4: [L188-L194] this paragraph does not belong in the discussion, it is more suited for the introduction.

Response 4: Paragraph was moved to Introduction.

Point 5: [L195] the study did not confirm ISOC as gold standard - this was the initial assumption.

Response 5: Sentence was amended and now says:

“Current study proposes a new, cheaper, smaller, and more portable device aiding commonly used isokinetic dynamometry due to ISOM’s strong validity and reliability found in present study for knee flexion and extension isometric strength assessment.”

Point 6: [L198-L200] in this section ISOC and ISOM were compared in terms of the design - it would be acceptable if more focus was put on the design of ISOM.

Response 6: Sentence was re-worded removing comparison with ISOC as follows:

“ISOM takes up relatively little deal of space and short set-up time and could therefore show its usefulness and spread its use among sports professionals.”

Point 7: [L210] "probably" is a little awkward.

Response 7: “probably” was removed.

Point 8: some sentences could be rewritten.

Response 8: We thank expert reviewer for his/her suggestion. English language and style were professionally revised by an academic proofreading service.

We hope that the manuscript has now reached the standard necessary for formal acceptance endorsement in International Journal of Environmental Research and Public Health.

We look forward to hearing from you.

Best regards

Reviewer 2 Report

In this study the authors aim was to investigate concurrent validity and reliability of an isometric dynamometer prototype. This prototype has been applied to assess knee flexion and extension isometric strength. The authors remark the benefits of the portability and low cost of the system, and it is the main value of the proposal. However, this paper lacks relevant novelty and originality because measuring isometric muscular strength with specific load cells has been reported in many previous experiments (e.g. Pua et al., 2018; Mesquita et al., 2019; Smithi et al., 2012; Bellar et al., 2015; Myers et al., 2015).

Another important limitation of this paper, mentioned also by the authors,  is the reduced sample size. The variability of strength values across participants is a great handicap, especially when their conclusions rely in the absence of differences (due also by the high dispersion of the data) and not in similarities.

Additionally, the methodology of this study must be carefully improved, specifically regarding the statistics and methods applied. Authors must provide more information about statistical procedures used, values of validity, similarities (instead of differences) in the measures compared to the “gold standard”. They must increase sample and perform repeated measures before conclude validity and reliability of the system.

Specific comments:

Page 2, line 47-48 It is not planned to compare the results to other portable systems like handheld dynamometer.

Page 2, line 65. The results from gold standard has to be compared to not only find differences but find similarities. Repeated measures on the same participants are strongly recommended to compare the results.

Page 2, line 71. The sample is reduced and all around 16 years old. Larger sample and distributed in a wider ranges of age and strength capabilities is recommended to validate this kind of system.

Page 2. Line 75-77. The inclusion criteria do not seem relevant to the aim of the study. It seems to be just the description of the sample. No rationale has introduced inclusion criteria.

Page 3, line 89-91. Deeper explanation of measurement is needed.

Page 3, line 116. This statement is not relevant in methods section.

Page 3, line 121. How was the position chosen? What heights are available?

Page 3. Line 122. How many antero-posterior positions? More information is recommended.

Page 3, line 125. Could the foot lie on the support plan? Did you measure the force applied to it?

Page 3, line 128. You comment that Due to ISOM novelty, it took an appropriate time to study its manual and practice enough with ISOM before starting participants testing. Were the experimenter practiced and instructed to measure? How many experimenters took part in the measurement? Did you analyze the inter-subject reliability of the procedure?

Page 3, line 129. Mechanical ISOM signals were recorded and digitalized by a portable device  you must specify. Data about acquisition, transformation, and transduction procedures are needed.

Page 4, line 134. Please provide information about the force values provided procedure.

Page 4, line 136. ISOC provide values of N.m instead of N. Please provide how did you calculate N by isokinetic system.

Page 4, line 138. The statistical method must be carefully detailed

Page 4, line 141. How did you calculate differences? Did you use repeated measures? Please include F values, DoF, effect size (in case you did an ANOVA).

Page 4, line 150. Did you find differences in left/right extension/flexion? Were the instruments capable to detect that differences? This experiment lacks relevant information for a validity experiment.

Page 5, line 167-168. Le reliability of ISOM and ISOC were not compared to support this statement.

Page 6, line 184-187. Handheld dynamometry have also demonstrated high reliability in knee isometric strength assessment. It is not clear how this study support or not the use of other typical load cells or handheld dynamometry.

Page 6, line 188. In the discussion section, the commentaries about the benefits of isometric strength for several sports do not add any support to the aim of this study, the validation a new protocol or instrument.

References

Pua YH, Poon CL, Seah FJ, et al. Comparative performance of isometric and isotonic quadriceps strength testing in total knee arthroplasty. Musculoskelet Sci Pract. 2018;37:17‐19. doi:10.1016/j.msksp.2018.06.008

Mesquita MMA, Santos MS, Vasconcelos ABS, et al. Reliability of a Test for Assessment of Isometric Trunk Muscle Strength in Elderly Women. J Aging Res. 2019;2019:9061839. Published 2019 Jul 1. doi:10.1155/2019/9061839

Smith BI, Docherty CL, Simon J, Klossner J, Schrader J. Ankle strength and force sense after a progressive, 6-week strength-training program in people with functional ankle instability. J Athl Train. 2012;47(3):282‐288. doi:10.4085/1062-6050-47.3.06

Bellar D, Marcus L, Judge LW. Validation and Reliability of a Novel Test of Upper Body Isometric Strength. J Hum Kinet. 2015;47:189‐195. Published 2015 Oct 14. doi:10.1515/hukin-2015-0074

Myers NL, Toonstra JL, Smith JS, Padgett CA, Uhl TL. Sustained isometric shoulder contraction on muscular strength and endurance: a randomized clinical trial. Int J Sports Phys Ther. 2015;10(7):1015‐1025.

Author Response

Response to Reviewer 2 Comments

Does the introduction provide sufficient background and include all relevant references?

(x) Must be improved

Please, read below comments to specific points.

Is the research design appropriate?

(x) Must be improved

Please, read below comments to specific points.

Are the methods adequately described?

(x) Must be improved

Please, read below comments to specific points.

Are the results clearly presented?

(x) Must be improved

Please, read below comments to specific points.

Are the conclusions supported by the results?

(x) Must be improved

Please, read below comments to specific points.

Point 1: this paper lacks relevant novelty and originality because measuring isometric muscular strength with specific load cells has been reported in many previous experiments (e.g. Pua et al., 2018; Mesquita et al., 2019; Smithi et al., 2012; Bellar et al., 2015; Myers et al., 2015).

Response 1: We thank expert reviewer for his/her suggestion. This paper’s novelty is represented by assessing a (patent pending) portable isokinetic dynamometer prototype. References to custom-built load (Smith et al., 2012; Bellar et al., 2015; Pua et al., 2018; and Mesquita et al., 2019) and commercial cells (Myers et al., 2015) were added.

Point 2: Another important limitation of this paper, mentioned also by the authors, is the reduced sample size. The variability of strength values across participants is a great handicap, especially when their conclusions rely in the absence of differences (due also by the high dispersion of the data) and not in similarities.

Response 2: Small sample size limitation was emphasised as follows:

“Variability of strength values affects particularly studies like present one showing absence of differences between measurement taken with different devices.”

Point 3: Additionally, the methodology of this study must be carefully improved, specifically regarding the statistics and methods applied. Authors must provide more information about statistical procedures used. They must perform repeated measures before concluding reliability of the system.

Response 3: Indeed, measurements were performed twice for reliability assessment’s aim. Reliability was analysed with the intraclass correlation coefficient. Detail was added to 2. Materials and Methods.

Point 4: Page 2, line 47-48 It is not planned to compare the results to other portable systems like handheld dynamometer.

Response 4: Whole paragraph was removed.

Point 5: Page 2, line 65. Repeated measures on the same participants are strongly recommended to compare the results.

Response 5: Please, read response 3.

Point 6: Page 2, line 71. The sample is reduced and all around 16 years old. Larger sample and distributed in a wider ranges of age and strength capabilities is recommended to validate this kind of system.

Response 6: Sample’s 16-year age was acknowledged within limitations as follows:

“Furthermore, sample is about 16 yrs old, whereas there may be age-related differences.”

Point 7: Page 2. Line 75-77. The inclusion criteria do not seem relevant to the aim of the study. It seems to be just the description of the sample. No rationale has introduced inclusion criteria.

Response 7: Sentence was changed as follows:

“All participants had following features: 1) participation in at least 85% of the training sessions; 2) possession a valid sport medical certification; 3) no injuries in the last year; and 4) no consumption of alcohol or drugs.”

Point 8: Page 3, line 89-91. Deeper explanation of measurement is needed.

Response 8: Sentence was split and re-worded.

Point 9: Page 3, line 116. This statement is not relevant in methods section.

Response 9: Sentence was removed.

Point 10: Page 3, line 121. How was the position chosen? What heights are available? Page 3. Line 122. How many antero-posterior positions? More information is recommended.

Response 10: Information was added as follows:

“Device has a removable custom bracket (Element A, Figures 1 and 2) that can be placed at five different vertical heights (from seat height down to about 70 cm below that, for different lower limbs lengths), different antero-posterior positions (continuous regulation from seat forward edge vertical projection up to 13 cm forward, for different knees angles), and pointing backwards (for flexion) or forwards (for extension). Element A was differently placed vertically and horizontally according to each participant’s shank length.”

Point 11: Page 3, line 125. Could the foot lie on the support plan? Did you measure the force applied to it?

Response 11: Information was added as follows:

“Structure is completed by a support plate for foot, where the foot was softly placed on without weighing significantly on it.”

Point 12: Page 3, line 128. You comment that Due to ISOM novelty, it took an appropriate time to study its manual and practice enough with ISOM before starting participants testing. Were the experimenter practiced and instructed to measure? How many experimenters took part in the measurement? Did you analyze the inter-subject reliability of the procedure?

Response 12: Information was added as follows:

“Author LR adjusted setting (set-up time about 5 min), RD took measures (included anthropometric ones), and JP supervised all operations.”

Point 13: Page 3, line 129. Mechanical ISOM signals were recorded and digitalized by a portable device you must specify. Data about acquisition, transformation, and transduction procedures are needed.

Response 13: Information on used devices (“H3-C3-750, Zemic Europe…+ FC5k, Axis…), force recording (“750 kg, 1000 Hz”) and processing (“ISOM strain gauge-provided force values in kN…) are already there in paragraph.

Point 14: Page 4, line 134. Please provide information about the force values provided procedure.

Response 14: I regret to say I don’t understand the request.

Point 15: Page 4, line 136. ISOC provide values of N.m instead of N. Please provide how did you calculate N by isokinetic system.

Response 15: Used ISOC software provides an “isometry” (condition) option, as it was the case. Namely, when “isometry” is flagged, ISOC provides force [N] values.

Point 16: Page 4, line 138. The statistical method must be carefully detailed.

Response 16: Information was added as follows:

“Statistical analysis (two-tailed paired samples Student t-test, n=17 for each of the four sides/joints movements considered) was performed using SPSS Version 17 (IBM, Chicago, IL, USA).”

Point 17: Page 4, line 141. How did you calculate differences? Did you use repeated measures? Please include F values, DoF, effect size (in case you did an ANOVA).

Response 17: Sentence and Table 1 refer to validity. Table 1’s footer was changed as follows:

“Data are presented as average±standard deviation for both devices (ISOC and ISOM). Sig= two-tailed paired samples Student t-test significance, LOA=ISOC-ISOM difference 95% limits of agreement (N).”

Two-tailed paired samples Student t-test was used.

Point 18: Page 4, line 150. Did you find differences in left/right extension/flexion? Were the instruments capable to detect that differences? This experiment lacks relevant information for a validity experiment.

Response 18: Lack of dominant vs. non-dominant comparison was acknowledged within limitations as follows:

“Dominant vs. non-dominant side measurements were not investigated.

Point 19: Page 5, line 167-168. The reliability of ISOM and ISOC were not compared to support this statement.

Response 19: I apologize for the mistake. Reference to ISOC reliability assessment was removed throughout MS.

Point 20: Page 6, line 184-187. … handheld dynamometry have also demonstrated high reliability in knee isometric strength assessment. It is not clear how this study support or not the use of other typical load cells or handheld dynamometry.

Response 20: Sentence was removed.

Point 21: Page 6, line 188. In the discussion section, the commentaries about the benefits of isometric strength for several sports do not add any support to the aim of this study, the validation a new protocol or instrument.

Response 21: We thank expert reviewer for his/her suggestion. Paragraph was moved to Introduction.

We hope that the manuscript has now reached the standard necessary for formal acceptance endorsement in International Journal of Environmental Research and Public Health.

We look forward to hearing from you.

Best regards

Reviewer 3 Report

- The study design was appropriate to answer the aim, this study adds practical applications of device-specific pros are cheapness and portability, This could allow the device to be used in a variety of strength and conditioning settings to assess training interventions or detraining effects, etc….. 

- INSUFFICIENT literature review in the introduction and discussion

- There are numerous errors of “Tense and Grammar” throughout the manuscript. Therefore, diligent editing is in order to fix the ENGLISH language.

- The major concern with the manuscript is that the authors have performed validity and reliability and yet they have only used 17 participants. Such a small sample size is inappropriate for the purposes of your investigation. This issue is insurmountable until the authors collect more data and I would recommend that the authors pursue this path and consider data on a group with different characteristics.

- Some references are not recent such as 11, 20, and 23.

- There are no presented results for reliability in the results section.

- Line 147 (It was observed only a small ISOC-ISOM measurement difference) ???.

- Line 148: figure 4 has to be figure 3.

-lines 185 - 187: The authors thank their results in the current study and I see it's not ethical. Please, avoid these sentences in the manuscript.

- The authors conclude that the cheaper device is practical and useful as a measuring tool for sports players. Is this enough with such small samples ???   

Author Response

Response to Reviewer 3 Comments

English language and style

(x) Moderate English changes required

English language and style were professionally revised by an academic proofreading service.

Does the introduction provide sufficient background and include all relevant references?

(x) Can be improved

Please, read below comments to specific points.

Is the research design appropriate?

(x) Can be improved

Please, read below comments to specific points.

Are the methods adequately described?

(x) Can be improved

Please, read below comments to specific points.

Are the results clearly presented?

(x) Can be improved

Please, read below comments to specific points.

Are the conclusions supported by the results?

(x) Can be improved

Please, read below comments to specific points.

Point 1: INSUFFICIENT literature review in the introduction and discussion.

Response 1: We thank expert reviewer for his/her suggestion. References were added, moved from Discussion to Introduction, and removed throughout Introduction and Discussion.

Point 2: There are numerous errors of “Tense and Grammar” throughout the manuscript. Therefore, diligent editing is in order to fix the ENGLISH language.

Response 2: English language and style were professionally revised by an academic proofreading service.

Point 3: The major concern with the manuscript is that the authors have performed validity and reliability and yet they have only used 17 participants. Such a small sample size is inappropriate for the purposes of your investigation. This issue is insurmountable until the authors collect more data and I would recommend that the authors pursue this path and consider data on a group with different characteristics.

Response 3: Small sample size limitation was emphasised as follows:

“Variability of strength values affects particularly studies like present one showing absence of differences between measurement taken with different devices.”

Sample’s 16-year age was acknowledged within limitations as follows:

“Furthermore, sample is about 16 yrs old, whereas there may be age-related differences.”

Point 4: Some references are not recent such as 11, 20, and 23.

Response 4: Reference 20 (now 19) is functional to Bland-Altman test use. Newer references were added.

Point 5: There are no presented results for reliability in the results section.

Response 5: ISOM reliability was analyzed with the intraclass correlation coefficient (ICC). ISOM ICCs were presented in 3.2. Reliability 3. Results subsection.

Point 6: Line 147 (It was observed only a small ISOC-ISOM measurement difference)???

Response 6: Sentence was re-worded as follows:

“Therefore, it was observed only a small (i.e., 3-6%; Table 1) difference between measurements with ISOC and measurements with ISOM, as illustrated in peak isometric forces Bland-Altman plots as well (Figure 3).”

Point 7: Line 148: figure 4 has to be figure 3.

Response 7: I apologize for the mistake. Yet, another Figure 3 was added, therefore Figure 4 remains Figure 4.

Point 8: lines 185 - 187: The authors thank their results in the current study and I see it's not ethical. Please, avoid these sentences in the manuscript.

Response 8: Sentence was re-worded as follows:

“Current study shows that here investigated prototype provides reliable results to assess knee flexors and extensors isometric strength and this supports its use.”

Point 9: The authors conclude that the cheaper device is practical and useful as a measuring tool for sports players. Is this enough with such small samples???

Response 9: We thank expert reviewer for his/her suggestion. Sentence was “smoothed” as follows:

“Net of above acknowledged present study’s limitations, sports professionals could therefore consider the possibility to use this dynamometer to assess isometric strength even during training process under field conditions.”

We hope that the manuscript has now reached the standard necessary for formal acceptance endorsement in International Journal of Environmental Research and Public Health.

We look forward to hearing from you.

Best regards

Reviewer 4 Report

Comments are added to the attached pdf file. 

Author Response

Response to Reviewer 4 Comments

English language and style

(x) Extensive editing of English language and style required

English language and style were professionally revised by an academic proofreading service.

Does the introduction provide sufficient background and include all relevant references?

(x) Must be improved

Please, read below comments to specific points.

Is the research design appropriate?

(x) Can be improved

Please, read below comments to specific points.

Are the methods adequately described?

(x) Must be improved

Please, read below comments to specific points.

Are the results clearly presented?

(x) Must be improved

Please, read below comments to specific points.

Are the conclusions supported by the results?

(x) Must be improved

Please, read below comments to specific points.

Point 1: (original review *.pdf, line 29) ICC 0.879÷0.990.

Response 1: We thank expert reviewer for his/her suggestion. “÷” was changed to “-” throughout MS.

Point 2: (l30) Be objective in the comparison, differences in terms of what?

Response 2: Sentence was removed.

Point 3: (page 3 bottom) Show the load cell/other device that were used in both ISOC and ISOM to measure the force in close frame picture.

Figure 1 and 2 are not informative.

Response 3: ISOC is a known commercial isokinetic dynamometer. Its load cell is embedded within the device. A figure with ISOM’s load cell was added. I do believe Figures 1 and 2 are informative, because they not only describe ISOM, but even show participant’s posture on dynamometer.

Point 4: (p4 top) There are many factors such as velocity, calibration of dynamometer, and etc that can affect the results and the author should clarify them. Did they consider them in their tests if not justify that.

Response 4: Lack of consideration regarding above aspects was added to limitations as follows:

“Furthermore, dynamometers calibrations and joints movements speeds were not considered, but may affect devices comparison and should be matter for further studies.”

Point 5: (l134) Why you did not consider torque measurement?

Response 5: Not considering isometric torque measurement was added to limitations as follows:

“Besides, isometric torques measurements were not explicitly considered.”

Point 6: (l139) Explain what statistical model was used, number of data points.

The numbers showed in this table were measured from whom? age, height, body mass?

Response 6: Information was added as follows:

“Author LR adjusted setting, RD took measures (included anthropometric ones), and JP supervised all operations… Statistical analysis (two-tailed paired samples Student t-test, n=17 for each of the four sides/joints movements considered) was performed using SPSS Version 17 (IBM, Chicago, IL, USA).”

Point 7: (Table 1) Explain LOA and how it was measured\calculated.

Response 7: Information was added as follows:

“Data are presented as average±standard deviation for both devices (ISOC and ISOM). Sig= two-tailed paired samples Student t-test significance, LOA=ISOC-ISOM difference 95% limits of agreement (N).”

Point 8: (l195) This paragraph should be in your introduction not discussion.

Response 8: Paragraph was moved to Introduction.

Point 9: This comparison is very subjective. Be specific and quantify the space and time set-up.

Response 9: We thank expert reviewer for his/her suggestion. Comparison with ISOC was removed and information was added as follows:

“Here assessed portable-with-seat dynamometer prototype (Isometric-Bench, Rome, Italy; ISOM; Figure 1) has a metallic steel structure, maximum size about 110 cm, and assembly time about 30 min (Figures 1 and 2)… Author LR adjusted setting (set-up time about 5 min), RD took measures (included anthropometric ones), and JP supervised all operations… ISOM takes up relatively little deal of space and short set-up time and could therefore show its usefulness and spread its use among sports professionals.”

We hope that the manuscript has now reached the standard necessary for formal acceptance endorsement in International Journal of Environmental Research and Public Health.

We look forward to hearing from you.

Best regards

Reviewer 5 Report

The authors present an interesting experimental study of comparison the commercial isokinetic dynamometer with the isometric dynamometer prototype. Although the topic is very hot, the presentation fails to provide the reader a clear view on the experimental procedure followed and the results obtained and thus to value the work in relation to the current state of the art in the field.
Apart of them the manuscript must be remarked in several aspects:
1.        In the abstract the prototype is called “isokinetic dynamometer prototype”. It is not true, but this opinion is difficult to confirm because the prototype is presented very generally. The In the abstract the prototype is called “isokinetic dynamometer prototype”. It is not true, but this opinion is difficult to confirm because the prototype is presented very generally. The manuscript suggest that is concern isometric assesments during  strength of lower limb.manuscript suggest that is concern isometric assessments during  strength of lower limb.
2.        Figure 2 is not enough to showing how the prototype works. The force measurement method is not presented.
3.        The question is: How is the force measured? It is not clearly that the weight of elements and person has not influence on the measure process.
4.        The manuscript has not include any information about the measurement conditions for both equipments. They are the same?.  The reader is not informed about the arm of forces (are the same or not),  patient position etc.
5.        In paragraph 3.1 the results including range +/-. What is it mean? It is the error or range given in several measurements. The measurement schedule is not clearly described.
6.        The measurement error given form the sensor and A/D quality is not analysed and compared. The cost of both equipments was raised but technical properties was not clearly compared.
7.        Table 1 not included correct units for parameters such as Sig, diff, LOA. The Sig parameter is not described!. The LOA is known but the shortcut should be described. These parameters should be comment in the text.
8.        The graphs on figure 3 have not units.
9.        The references was prepared quite superficially. The latest item on the description of similar problems is one of 2018. The authors do not refer to the results obtained so far in the literature, which makes it difficult to assess whether the tests were carried out correctly.

The article should be remarked and reviewed again.

Author Response

Response to Reviewer 5 Comments

Does the introduction provide sufficient background and include all relevant references?

(x) Must be improved

Please, read below comments to specific points.

Is the research design appropriate?

(x) Must be improved

Please, read below comments to specific points.

Are the methods adequately described?

(x) Must be improved

Please, read below comments to specific points.

Are the results clearly presented?

(x) Must be improved

Please, read below comments to specific points.

Are the conclusions supported by the results?

(x) Must be improved

Please, read below comments to specific points.

Point 1: In the abstract the prototype is called “isokinetic dynamometer prototype”. It is not true, but this opinion is difficult to confirm, because the prototype is presented very generally.

Response 1: We thank expert reviewer for his/her suggestion. Several further details were added regarding prototype’s description throughout MS.

Point 2: Figure 2 is not enough to show how the prototype works. The force measurement method is not presented. The question is: How is the force measured? It is not clear that the weight of elements and person has not influence on the measure process.

Response 2: Several further details were added regarding how prototype works throughout MS. Under, e.g., lower limb setting, load cell measures horizontal (or “tangential”) knee flexion or extension isometric force. This was specified as follows:

“Namely, load cell measures horizontal (or “tangential”) knee flexion or extension isometric force.”

No prototype’s elements or participant (vertical) weights interfere with load cell’s (horizontal) measurement.

Point 3: The manuscript has not included any information about the measurement conditions for both equipments. Are they the same? The reader is not informed about the arm of forces (are the same or not?), patient position, etc.

Response 3: Measurement conditions were the same with both dynamometers. Further details were added as follows:

“Particular care was taken to keep participant’s same posture (and resulting knee moment arms) with both dynamometers.”

Point 4: In paragraph 3.1 the results include range +/-. What is it mean?

Response 4: “±” special (mathematics) character means “plus or minus” and specifies result is given as “average±standard deviation”. This was made more explicitly as follows:

“ISOC’s left and right knee extension peak isometric forces resulted (average±standard deviation) 415.40±115.15 and 402.81±41.73 N, respectively, compared to corresponding ISOM’s values 394.73±123.28 and 379.46±50.78 N (+20.64 N/+5.1% and +23.39 N+5.9%).”

Point 5: The measurement error given from the sensor and A/D quality is not analysed and compared. The costs of both equipments were considered, but technical properties were not clearly compared.

Response 5: ISOM validity was assessed compared with ISOC gold standard. ISOM reliability was assessed by means of test-retest comparison. A/D quality assessment went beyond study’s aim. Dynamometers’ technical properties comparison went beyond study’s aim.

Point 6: Table 1 not included correct units for parameters such as Sig, diff, LOA. The Sig parameter is not described! The LOA is known, but the shortcut should be described. These parameters should be comment in the text.

Response 6: Table 1’s footer was changed as follows:

“Data are presented as average±standard deviation for both devices (ISOC and ISOM). Sig= two-tailed paired samples Student t-test significance, LOA=ISOC-ISOM difference 95% limits of agreement (N).”

Table 1 was commented in Results.

Point 7: The graphs on figure 3 have not units.

Response 7: Figure 4 (previously Figure 3) caption was changed as follows:

“Figure 4. Knee extension and flexion peak isometric forces (N) Bland–Altman plots with both devices.”

Point 8: The references were prepared quite superficially. The latest item on the description of similar problems is one of 2018. The authors do not refer to the results obtained so far in the literature, which makes it difficult to assess whether the tests were carried out correctly.

Response 8: We thank expert reviewer for his/her suggestion. References were added, moved from Discussion to Introduction, and removed throughout Introduction and Discussion.

We hope that the manuscript has now reached the standard necessary for formal acceptance endorsement in International Journal of Environmental Research and Public Health.

We look forward to hearing from you.

Best regards

Round 2

Reviewer 1 Report

Thank you for responding to my comments.

-> this is probably from the pdf conversion, but Fig.3 is too large - not proportional to other figures in the manuscript.

Author Response

Response to Reviewer 1 Comments

Point 1: -> this is probably from the pdf conversion, but Fig. 3 is too large - not proportional to other figures in the manuscript.

Response 1: We thank expert reviewer for his/her comment (and overall approval). Figure 3 was removed.

We hope that the manuscript has now reached the standard necessary for formal acceptance endorsement in International Journal of Environmental Research and Public Health.

We look forward to hearing from you.

Best regards

Reviewer 2 Report

I want to thank the authors their effort in improving the manuscript. I understand that carry out an experiment is a hard work and not always with the appropriate reward. Nevertheless, the main problem about this paper is still its relevancy, novelty and originality. As I mentioned earlier, measuring isometric muscular strength with specific load cells has been reported in many previous experiments. Authors support the novelty by a patent pending of an isokinetic dynamometer prototype. One patent could not be necessarily relevant, indeed there are many of them and with different purposes, not always relevant. In this manuscript, I assessed the relevancy of the study and the study still lacks relevancy. Indeed, an isokinetic prototype is not presented but an isometric protocol.

Other limitations of this paper are still mentioned also by the authors, like the reduced sample size and its characteristics, the lack of other comparisons and validity of the experiment. It is good to mention limitations but, sometimes, limitations of a study determine its relevance. Many sentences that would be explained due its relevancy. Like that related to reliability of ISOM and ISOC comparison (or how this study support or not the use of other typical load cells or handheld dynamometry), have been removed without an explanation.

The authors have provided more information about the statistics. They performed a paired t-test, but as I mentioned in the previous review, no difference is not the same that similarities.

Regarding the comment about how the foot lie on the support plan and the possible force applied, they only mention that the force applied was non significant, without any measurement of that.

Regarding the appropriate time to study and practice with ISOM before starting participants testing, authors do not respond about the instruction and more important, about the inter-subject reliability of the procedure.

Regarding the acquisition of mechanical ISOM signals, there are still no information about data transduction and A/D transformation, force calculation, filtering etc. A strain gauge do not provide information of force directly, it has to be transduced.

Summarizing, the manuscript still lacks relevancy and originality, maintain some methodological flaws, and do not present an alternative isokinetic prototype but an isometric protocol. I want to thank the author for their effort but, to my concern, the manuscript do not reach the conditions for acceptance in International Journal of Environmental Research and Public Health.

Author Response

Point 1: As I mentioned earlier, measuring isometric muscular strength with specific load cells has been reported in many previous experiments. Authors support the novelty by a patent pending of an isokinetic dynamometer prototype. Indeed, an isokinetic prototype is not presented but an isometric protocol.

Response 1: We thank expert reviewer for his/her comment. Differently from previous papers, current MS refers to a ready-to-use isometric force measurement system. I mean that here assessed system is not made by a specific load cell only, rather there is load cell, there is seat, there is plenty of adjustable settings for measuring different joint isometric forces… In the meantime, patent was obtained. I agree an isometric dynamometer is presented here. “isokinetic” term misuse was amended throughout MS.

Point 2: Like that related to reliability of ISOM and ISOC comparison (or how this study support or not the use of other typical load cells or handheld dynamometry), have been removed without an explanation.

Response 2: Reference to comparison between ISOM and ISOC reliabilities was removed, because ISOC reliability was not assessed. Reference to handheld dynamometry was removed, because handheld dynamometry was not object of investigation of current research.

Point 3: Regarding the comment about how the foot lies on the support plan and the possible force applied, they only mention that the force applied was non significant, without any measurement of that.

Response 3: Foot support plate can be placed at different heights. By placing plate correctly, I believe vertical force is negligible compared with horizontal (or “tangential”) one. Detail was added as follows:

“Foot support plate can be placed at any different vertical heights (from seat height down to about 80 cm below that, for different lower limbs lengths).”

Point 4: Regarding the appropriate time to study and practice with ISOM before starting participants testing, authors do not respond about the instruction and more important, about the inter-subject reliability of the procedure.

Response 4: Inter-subject reliability was not assessed, because only one author (RD) took measures.

Point 5: Regarding the acquisition of mechanical ISOM signals, there are still no information about data transduction and A/D transformation, force calculation, filtering etc.

Response 5: We thank expert reviewer for his/her comment. No filtering was applied. Details were added as follows:

“Mechanical ISOM signals were recorded and digitalized by a portable device (5 kN full scale; 1 N reading graduation; 24-bit resolution; FC5k, Axis, Gdańsk, Poland).”

We hope that the manuscript has now reached the standard necessary for formal acceptance endorsement in International Journal of Environmental Research and Public Health.

We look forward to hearing from you.

Best regards

Reviewer 3 Report

Dear authors,

Thank you for your response regarding my comments. 

Author Response

Thanks for your constructive feedback.

Reviewer 4 Report

Please see the 

Point 3: (page 3 bottom) Show the load cell/other device that were used in both ISOC and ISOM to measure the force in close frame picture.

Figure 1 and 2 are not informative.

Response 3: ISOC is a known commercial isokinetic dynamometer. Its load cell is embedded within the device. A figure with ISOM’s load cell was added. I do believe Figures 1 and 2 are informative, because they not only describe ISOM, but even show participant’s posture on dynamometer.

Reviewer #2nd round: You can show the ISOM and posture of the participants in one picture not four. My point was to put the picture of ISOC with the posture of participants as well.

Please do remove the Fig 3. It is not necessary If the load cell embedded in both devices, just explain the mechanism.

Point 4: (p4 top) There are many factors such as velocity, calibration of dynamometer, and etc that can affect the results and the author should clarify them. Did they consider them in their tests if not justify that.

Response 4: Lack of consideration regarding above aspects was added to limitations as follows:

“Furthermore, dynamometers calibrations and joints movements speeds were not considered, but may affect devices comparison and should be matter for further studies.”

Reviewer #2nd round:Please explain how they can affect the results; do you think it is important to consider?

Point 6: (l139) Explain what statistical model was used, number of data points.

The numbers showed in this table were measured from whom? age, height, body mass?

Response 6: Information was added as follows:

“Author LR adjusted setting, RD took measures (included anthropometric ones), and JP supervised all operations… Statistical analysis (two-tailed paired samples Student t-test, n=17 for each of the four sides/joints movements considered) was performed using SPSS Version 17 (IBM, Chicago, IL, USA).”

Reviewer #2nd round: Please do remove the highlighted part from the text. I did ask to mention the contribution of the authors in the project. What I meant was to explain the measurements. For example, instead of showing only the average you can present the 17 data points in the column bar based on BMI and show the maximum, minimum, for extension and flexion.

Please explicitly explain that considering all these limitations that you have in your study how you conclude that this device is reliable? 

Author Response

Response to Reviewer 4 Comments

Point 1: You can show the ISOM and posture of the participants in one picture not four.

Response 1: We thank expert reviewer for his/her suggestion. Figure 2 was removed.

Point 2: Please do remove the Fig 3. It is not necessary If the load cell embedded in both devices, just explain the mechanism.

Response 2: Figure 3 was removed. Specification was added as follows:

“A load cell (750 kg, 1000 Hz; H3-C3/C4-750kg-3B, Zemic Europe, Etten Leur, Netherland) is connected (i.e., embedded in) to Element A, which in turn connects load cell to a metal plate perfectly in contact with tested lower limb skin (pressure point).”

Point 3: There are many factors such as velocity, calibration of dynamometer, etc. that can affect the results and the author should clarify them. Please explain how they can affect the results; do you think it is important to consider?

Response 3: Sentence was split and re-phrased as follows:

“Furthermore, dynamometers calibrations were not considered, but may affect devices comparison. Calibration complexity, duration, and cost are factors to be taken into consideration and should be matter for further studies.”

Point 4: For example, instead of showing only the average you can present the 17 data points in the column bar based on BMI and show the maximum, minimum, for extension and flexion.

Response 4: I regret to write I do not understand well enough your point. Results text reports average±standard deviation ISOC’s and ISOM’s left and right knee extension and flexion peak isometric forces with average ISOC-ISOM absolute and percent differences. Table 1 reports two-tailed paired samples Student t-test significances and ISOC-ISOM absolute difference 95% limits of agreement (viz. 95% of 17 measures lie within them), as well. Its caption was further detailed as follows:

“Table 1. Knee extension and flexion peak isometric forces with both devices (n=17).”

Figure 2 shows all (i.e., 100% of [double] 17 measures for each side/joint contraction) ISOC-ISOM absolute differences as Bland-Altman plots. Its caption was further detailed as follows:

“Figure 2. Knee extension and flexion peak isometric forces (N) Bland–Altman plots with both devices (n=17). ExtLeftISOC=left knee extension with ISOC, ExtLeftISOM=left knee extension with ISOM, ExtRightISOC=right knee extension with ISOC, ExtRightISOM=right knee extension with ISOM, FlexLeftISOC=left knee flexion with ISOC, FlexLeftISOM=left knee flexion with ISOM, FlexRightISOC=right knee flexion with ISOC, FlexRightISOM=right knee flexion with ISOM”

Point 5: Please explicitly explain that considering all these limitations that you have in your study how you conclude that this device is reliable?

Response 5: We thank expert reviewer for his/her suggestion. Following sentence was added:

“Overall, authors believe ISOM represents an effective device for sports professionals aiming at assessing knee isometric force.”

We hope that the manuscript has now reached the standard necessary for formal acceptance endorsement in International Journal of Environmental Research and Public Health.

We look forward to hearing from you.

Best regards

Reviewer 5 Report

In my opinion the figure 3 is unnecessary. Does not carry any information. There is still no description of how the sensor is loaded. Description in line 123 is not enough. I suggest you delete this photo. I do not understand the changes made in the literature. There have been changes that appear in the text as the same quotes.

Author Response

Point 1: In my opinion the figure 3 is unnecessary.

Response 1: We thank expert reviewer for his/her suggestion. Figure 3 was removed.

Point 2: There is still no description of how the sensor is loaded.

Response 2: Further details about how load cell works and foot support plate were added as follows:

“Element A was differently placed vertically and horizontally according to each participant’s shank length. A load cell (750 kg, 1000 Hz; H3-C3/C4-750kg-3B, Zemic Europe, Etten Leur, Netherland) is connected (i.e., embedded in) to Element A, which in turn connects load cell to a metal plate perfectly in contact with tested lower limb skin (pressure point). Namely, load cell measures horizontal (or “tangential”) knee flexion or extension isometric force applied against it. … Foot support plate can be placed at any different vertical heights (from seat height down to about 80 cm below that, for different lower limbs lengths).

Point 3: I do not understand the changes made in the literature. There have been changes that appear in the text as the same quotes.

Response 3: We thank expert reviewer for his/her suggestion. Following reference was added to Introduction:

  1. Lienhard, K., Lauermann, S.P., Schneider, D., Item-Glatthorn, J.F., Casartelli, N.C., Maffiuletti, N.A. Validity and reliability of isometric, isokinetic and isoinertial modalities for the assessment of quadriceps muscle strength in patients with total knee arthroplasty. J Electromyogr Kinesiol 2013, 23, 1283‐1288. DOI: 10.1016/j.jelekin.2013.09.004

When revising MS for first time, a whole paragraph with its references was moved from Discussion to Introduction because more suitable for it.

We hope that the manuscript has now reached the standard necessary for formal acceptance endorsement in International Journal of Environmental Research and Public Health.

We look forward to hearing from you.

Best regards